# Antidepressant discontinuation before or during pregnancy and risk of psychiatric emergency in Denmark: A population-based propensity score–matched cohort study

Xiaoqin Liu[1]*, Nina Molenaar[2], Esben Agerbo[1,3,4], Natalie C. Momen[1], Anna-Sophie Rommel[2], Angela Lupattelli[5], Veerle Bergink[2,6‡], Trine Munk-Olsen[1,7‡]

1 NCRR-The National Centre for Register-based Research, Aarhus University, Aarhus, Denmark,
2 Department of Psychiatry, Icahn School of Medicine at Mount Sinai, New York, New York, United States of America, 3 iPSYCH-Lundbeck Foundation Initiative for Integrative Psychiatric Research, Aarhus, Denmark,
4 CIRRAU-Centre for Integrated Register-based Research, Aarhus University, Aarhus, Denmark,
5 PharmacoEpidemiology and Drug Safety Research Group, Department of Pharmacy, and PharmaTox Strategic Research Initiative, Faculty of Mathematics and Natural Sciences, University of Oslo, Oslo, Norway,
6 Department of Psychiatry, Erasmus Medical Center, Rotterdam, the Netherlands, 7 Department of Clinical Research, University of Southern Denmark, Odense, Denmark

‡ These authors are joint senior authors on this work.
* lxq@econ.au.dk

**Data Availability Statement:** The study was based on Danish nationwide registers (https://econ.au.dk/

## Abstract

### Background

Women prescribed antidepressants face the dilemma of whether or not to continue their treatment during pregnancy. Currently, limited evidence is available on the efficacy of continuing versus discontinuing antidepressant treatment during pregnancy to aid their decision. We aimed to estimate whether antidepressant discontinuation before or during pregnancy was associated with an increased risk of psychiatric emergency (ascertained by psychiatric admission or emergency room visit), a proxy measure of severe exacerbation of symptoms/mental health crisis.

### Methods and findings

We carried out a propensity score–matched cohort study of women who gave birth to live-born singletons between January 1, 1997 and June 30, 2016 in Denmark and who redeemed an antidepressant prescription in the 90 days before the pregnancy, identified by Anatomical Therapeutic Chemical (ATC) code N06A. We constructed 2 matched cohorts, matching each woman who discontinued antidepressants before pregnancy ($N$ = 2,669) or during pregnancy ($N$ = 5,467) to one who continued antidepressants based on propensity scores. Maternal characteristics and variables related to disease severity were used to generate the propensity scores in logistic regression models. We estimated hazard ratios (HRs) of psychiatric emergency in the perinatal period (pregnancy and 6 months postpartum) using stratified Cox regression.

the-national-centre-for-register-based-research/
danish-registers and https://kea.au.dk/research/
registers-and-biobank/the-danish-national-health-
service-prescription-database), and the data are
not collected for a specific research project.
According to the Danish legislation, individual-level
data can be accessed only through secure servers,
Denmark Statistics, where download or export of
individual-level information is prohibited. Only
aggregated data can be shared to ensure complete
anonymity and protection of individuals included in
the studies. Permission to work on Danish
registers for research purposes can be granted to
individuals when a set of requirements are fulfilled,
including employment or affiliation with a Danish
research institution. To know more about how to
access the data, please contact the Centre for
Integrated Register-based Research at Aarhus
University (CIRRAU) at Aarhus University (https://
cirrau.au.dk). For any inquiries, please contact
Professor Carsten Bøcker Pedersen (cbp@econ.au.
dk) at the National Centre for Register-based
Research, Aarhus University, Denmark.

**Funding:** XL is supported by the European Union's
Horizon 2020 research and innovation programme
under the Marie Sklodowska-Curie grant
agreement No 891079. NM, NCM, AR, VB, and
TMO are supported by the National Institute of
Mental Health (NIMH) (R01MH122869). EA is
supported by iPSYCH, the Lundbeck Foundation
Initiative for Integrative Psychiatric Research
(R155-2014-1724). TMO is also supported by the
Lundbeck Foundation (R313-2019-569), AUFF
NOVA (AUFF-E 2016-9-25), and Fabrikant Vilhelm
Pedersen og Hustrus Legat. AL is supported by the
Norwegian Research Council (grant no. 288696).
The investigators conducted the research
independently. The funders of the study had no
role in study design, data analysis, data
interpretation, writing, or submission for
publication.

**Competing interests:** The authors have declared
that no competing interests exist.

**Abbreviations:** ATC, Anatomical Therapeutic
Chemical; CI, confidence interval; HR, hazard ratio;
MAOI, monoamine oxidase inhibitor; SNRI,
serotonin–norepinephrine reuptake inhibitor; SSRI,
selective serotonin reuptake inhibitor; STROBE,
Strengthening the Reporting of Observational
Studies in Epidemiology; TCA, tricyclic
antidepressant.

Psychiatric emergencies were observed in 76 women who discontinued antidepressants before pregnancy and 91 women who continued. There was no evidence of higher risk of psychiatric emergency among women who discontinued antidepressants before pregnancy (cumulative incidence: 2.9%, 95% confidence interval [CI]: 2.3% to 3.6% for discontinuation versus 3.4%, 95% CI: 2.8% to 4.2% for continuation; HR = 0.84, 95% CI: 0.61 to 1.16, $p$ = 0.298). Overall, 202 women who discontinued antidepressants during pregnancy and 156 who continued had psychiatric emergencies (cumulative incidence: 5.0%, 95% CI: 4.2% to 5.9% versus 3.7%, 95% CI: 3.1% to 4.5%). Antidepressant discontinuation during pregnancy was associated with increased risk of psychiatric emergency (HR = 1.25, 95% CI: 1.00 to 1.55, $p$ = 0.048). Study limitations include lack of information on indications for antidepressant treatment and reasons for discontinuing antidepressants.

### Conclusions

In this study, we found that discontinuing antidepressant medication during pregnancy (but not before) is associated with an apparent increased risk of psychiatric emergency compared to continuing treatment throughout pregnancy.

## Author summary

### Why was this study done?

- Antidepressants are the first-line pharmacological treatment options for moderate to severe depression and anxiety disorders in the perinatal period, but over 50% of women discontinue their antidepressants during pregnancy, possibly due to concerns about negative effects on the unborn child.

- Premature antidepressant discontinuation may have deleterious consequences, and limited evidence is available on the efficacy of continuous antidepressant treatment during pregnancy to advise any decision on continuing antidepressants or not.

### What did the researchers do and find?

- We conducted a population-based propensity score–matched cohort study utilizing data from Danish nationwide registers.

- We found that antidepressant discontinuation during pregnancy was associated with an increased risk of psychiatric emergency (hazard ratio [HR] 1.25, 95% confidence interval [CI] 1.00 to 1.55), compared to continuing antidepressant treatment during pregnancy.

### What do these findings mean?

- Our findings add to the limited evidence on the relationship between continued antidepressant treatment during pregnancy and perinatal psychiatric emergency.

- Even though we have controlled for various demographic and clinical factors, residual confounding by indications for antidepressant treatment and disease severity cannot be ruled out.

## Introduction

Depression and anxiety are the most common mental disorders, affecting millions of adults worldwide [1]. Antidepressants are the first-line agents for treating these disorders, and over the years, antidepressant prescriptions have increased significantly [2], as has long-term maintenance treatment with antidepressants to prevent relapse of psychiatric disorders [3]. Similarly, antidepressant use during pregnancy is common: Approximately 2% to 8% of pregnant women in Europe [4] and 8% to 13% in the United States of America [5] receive antidepressant prescriptions at some point in their pregnancy. However, concerns have been raised about offspring sequelae of in utero antidepressant exposure [6,7], and, consequently, more than 50% of women discontinue antidepressants during pregnancy [8].

Whether or not antidepressants can be discontinued safely before or during pregnancy is unclear. A limited number of studies have compared relapse risk in pregnant women who discontinued antidepressants with those who continued, with mixed results: Two studies suggested that discontinuing antidepressants increased relapse risk compared with continuing [9,10], and 2 did not demonstrate clear benefits of continuing antidepressant treatment in pregnancy [11,12], while another study reported reduced risk associated with antidepressant discontinuation [13]. The magnitude of the relative risk of relapse for antidepressant discontinuation versus continuation ranges from 0.45 to 8.79 [9–13]. A recent meta-analysis pooled these results and found a borderline increased relapse risk of depression during pregnancy for women who discontinued antidepressants (risk ratio = 1.74, 95% confidence interval [CI]: 0.97 to 3.10) [14]. However, all studies differed by study populations, depression severity, time of discontinuation, relapse assessment, and a high degree of heterogeneity between studies was reported ($I^2$ = 94%). Unsurprisingly, current guidelines on antidepressant treatment in the perinatal period give few specific recommendations [15].

Untreated or incompletely treated depression or anxiety during pregnancy is associated with poor maternal health and adverse health outcomes in offspring [16,17], or at its worst, suicide or infanticide in the perinatal period [18], and thus should be avoided, in particular, to prevent severe exacerbations requiring admission or emergency room visits. Women are more vulnerable to severe psychiatric episodes during the perinatal period than at any other point during their lives [19]. Earlier studies suggested that the risks of suicidal ideation and admissions are high following antidepressant discontinuation, but these studies were small and had selected patient populations [9,20]. In contrast, a large register-based study found lower prevalence of hospitalization late in pregnancy in women who discontinued (9%) versus women who continued antidepressants (17%) [13].

In the present large and nationally representative study, we aimed to evaluate the risk of psychiatric emergency, measured as psychiatric admissions or emergency room visits in a population of pregnant women who discontinued antidepressants before or during pregnancy. We did this by defining psychiatric emergency as a proxy of severe exacerbation of symptoms/mental health crisis. We were interested in 2 time periods: pregnancy and the first 6 months postpartum (the perinatal period), and we hypothesized that women who discontinued

treatment before or during pregnancy would be more likely to have a subsequent psychiatric emergency in the perinatal period than women who continued antidepressants.

## Methods

This study is reported as per the Strengthening the Reporting of Observational Studies in Epidemiology (STROBE) guideline (S1 STROBE Checklist). There is no documented analysis plan associated with the study. We planned our analyses through detailed discussion between the authors and agreed on an outline for how the work would be carried out, as described in the Methods. One change to the preplanned analysis was to include a sensitivity analysis by including age at pregnancy and age at first affective disorder treatment as linear splines, implemented in response to peer reviewer comments.

### Study population

We conducted a population-based propensity score–matched cohort study utilizing data from Danish nationwide registers. A detailed description of registers used in this study can be found in S1 Text. Through the linkage of the Danish Medical Birth Registry and the Danish National Prescription Registry, we identified 23,189 women aged 18 years or older when they were pregnant and with pregnancies resulting in live-born singletons between January 1, 1997 and June 30, 2016, who redeemed an antidepressant prescription in the 90 days before the index pregnancy started (Fig 1), among whom 9,573 (41.3%) had a psychiatric diagnosis before pregnancy recorded in the Danish Psychiatric Central Research Register. The start of pregnancy was estimated by subtracting gestational age (primarily based on first or second trimester ultrasound scan) from birth date [21]. When no ultrasound data were available, the first day of the mother's last menstrual period was used. We included only the first pregnancy meeting the inclusion criteria.

### Exposure of interest: Antidepressant continuation during pregnancy

Information on antidepressant use was obtained from the Danish National Prescription Registry identified with the Anatomical Therapeutic Chemical (ATC) code of N06A. Antidepressants were categorized into 3 groups: selective serotonin reuptake inhibitors (SSRIs), serotonin–norepinephrine reuptake inhibitors (SNRIs), and tricyclic antidepressants [TCAs] or monoamine oxidase inhibitors [MAOIs]. The number of days' supply per prescription was calculated by multiplying the number of defined daily doses per packet by the number of packets dispensed. Prescriptions for the same antidepressants issued on the same day were counted as a single prescription and the days' supply combined [22].

We defined antidepressant continuation throughout pregnancy as continuous treatment from 3 months prior to until the end of the pregnancy based on the supply of antidepressants (number of days), allowing a 14-day grace period to account for missed doses [13]. If another prescription was not redeemed before the date when supply was expected to finish plus the 14-day grace period, the treatment was defined as discontinued on that date. Women who switched to other antidepressants within the date when the last supply finished plus the 14 days were considered as continuing treatment. Note, these definitions were applied to all individuals included in the study. We categorized all individuals who discontinued antidepressants according to time of discontinuation: before (within 90 days before conception) or during pregnancy. We also considered an alternative definition of antidepressant discontinuation, using a longer (30 days) grace period in the sensitivity analyses [23].

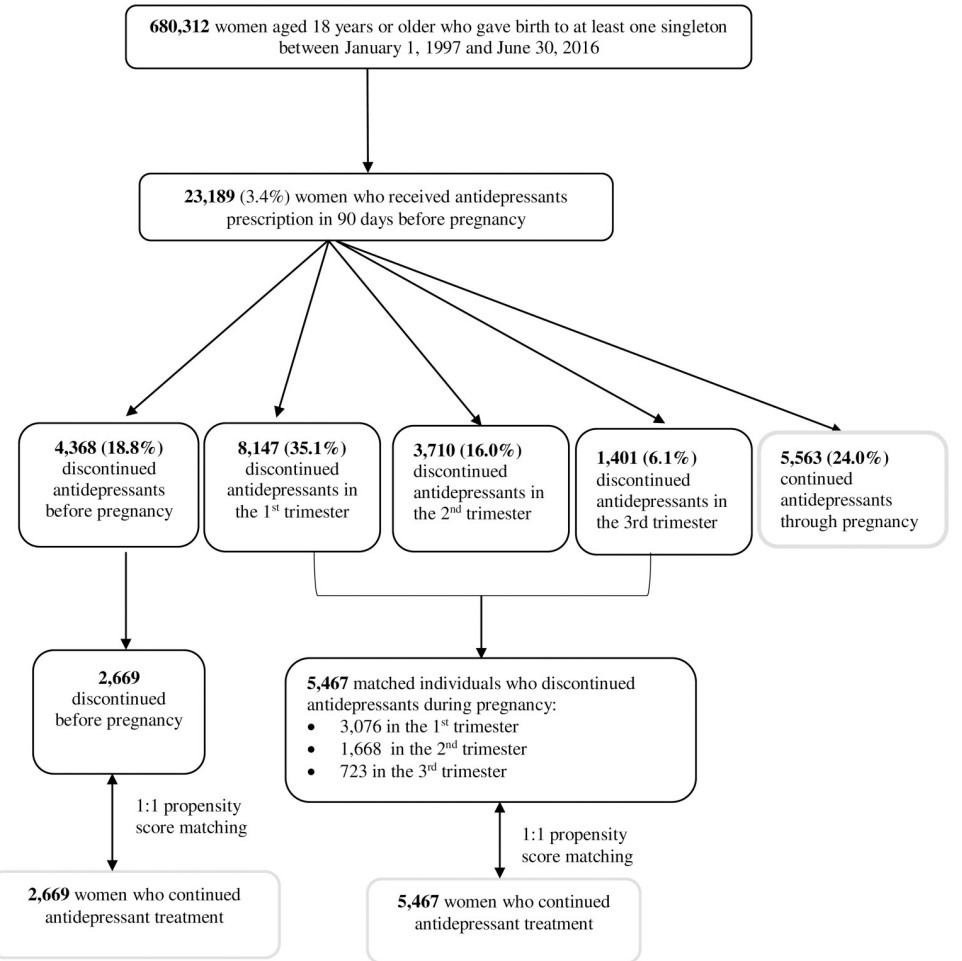

**Fig 1. Flowchart illustrating the identification of study population.**

### Propensity score–matched cohorts

Propensity score matching is commonly used to control for confounding in pharmacoepidemiological studies, especially when outcomes are rare, many confounders are present, and/or there are systematic differences in characteristic distribution between groups [24]. We calculated the exposure propensity score from the predicted probability of discontinuing antidepressants estimated in a logistic regression model containing all variables listed below, and then we matched each woman who discontinued antidepressants before pregnancy to one woman who continued antidepressants on propensity score using the nearest neighbor matching algorithm, within caliper widths of 0.1 without replacement. We similarly matched each woman who discontinued antidepressants during pregnancy to one woman who continued antidepressants (Fig 1). We chose caliper widths of 0.1 over the recommended 0.2 [25] based on the following considerations: (1) a tighter caliper leads to significantly reduced bias [26]; and (2) the increase in the number of matched pairs using caliper widths of 0.2 in our study was negligible.

The following variables considered potentially prognostically important for psychiatric emergency [24] were included to generate the propensity score:

1. Characteristics measured at the start of pregnancy: age (18 to 25, 26 to 29, 30 to 34, or ≥35 years), parity (first or second and above), marital status (married or cohabiting or single, divorced, or widowed), level of education (mandatory school comprising 9 school years or above mandatory school), age at the first affective episode (≤18, 19 to 25, 26 to 29, 30 to 34, or ≥35 years), history of suicide attempts (yes or no), diagnosed psychiatric disorders (substance abuse disorder, schizophrenia, bipolar disorder, depression, other mood disorders, neurotic, stress-related, and somatoform disorders, personality disorders, and other disorders; yes or no), the number of previous psychiatric emergency prior to 90 days before pregnancy, i.e., psychiatric admission or emergency room visit (0, 1, 2, 3 to 4, or ≥5), nonpsychiatric comorbidity scores (0, 1, or ≥2) based on the 19 conditions included in the Charlson comorbidity index [27]. The 19 conditions and Charlson comorbidity index score can be seen in S2 Text. We retrieved data on comorbidity from the Danish National Patient Register [28]. Age at first affective disorder was defined as the first hospital contact for the affective disorder (ICD-8 codes 296.x9, 298.09, 298.19, 300.49, 301.19, 300.x9, 305.x9, 305.68, and 307.99 excluding 296.89; ICD-10 codes F30 to F39 and F40 to F48) or first prescription redeemed for antidepressants (ATC code N06A) or anxiolytics (ATC code N05B), whichever occurred first. We also repeated our analyses by including age at pregnancy as linear splines with 4 knots and age at first affective disorder as linear splines with 5 knots at specified values based on age groups mentioned above to test the robustness of our results. Suicide attempts were defined as an admission or outpatient visit for self-harm identified from the Danish Psychiatric Central Research Register or the Danish National Patient Register [29]. Detailed criteria for defining suicide attempts can be found in S3 Text and ICD codes for subgroups of psychiatric disorders in S1 Table. We further included calendar year of the index pregnancy when calculating the propensity score to account for changes in prescribing patterns.

2. Variables related to the severity of disorders measured in the 90 days prior to pregnancy: classes of antidepressant treatment (SSRIs, SNRIs, and TCAs or MAOIs), psychiatric admission/emergency room visit (yes or no), and comedications (benzodiazepine, anxiolytics, antipsychotics, opioid, barbiturates, antiepileptics, and other hypnotics; yes or no). Detailed ATC codes for classes of antidepressants and comedications are shown in S2 and S3 Tables.

## Outcome of interest: Psychiatric emergency

Our primary outcome was first admission or emergency room visit with the main diagnosis of any mental disorders except for mental retardation (hereafter referred to as psychiatric emergency; the entire F chapter excluding F70 to F79 in ICD-10) during follow-up as a proxy for severe exacerbation of symptoms/mental health crisis. We also explored suicide attempts as an outcome. However, suicide attempts were rare, 0.4% to 0.5% in the matched cohorts, and therefore did not provide a meaningful outcome.

## Statistical analysis

For women who discontinued antidepressants, follow-up began on the date of discontinuation or the first day of pregnancy, whichever occurred later. Follow-up in the reference women (i.e., those who continued treatment) started at the same point in relation to the start of pregnancy as the woman they were matched to. For example, if an index woman discontinued antidepressants 50 days after pregnancy, follow-up commenced from 50 days after pregnancy for both this woman and her matched reference. For both the index and reference women, follow-

up ended at the earliest of the following: psychiatric inpatient, emergency room visit, death, emigration, or 6 months after childbirth. Fewer than 5 women emigrated or died in the matched cohort for antidepressant discontinuation before pregnancy; 12 women emigrated, and fewer than 5 women died in the matched cohort for antidepressant discontinuation during pregnancy by the end of follow-up. We employed a running line least squares smoothing technique [30] to smooth Kaplan–Meier curves to illustrate the cumulative incidence of psychiatric emergency following the start of follow-up by antidepressant continuation/discontinuation.

We calculated standardized differences to assess covariate balance before and after propensity score matching between groups; meaningful imbalances were defined by an absolute standardized difference of more than 0.1 [31]. Stratified Cox regression [32] was used to estimate the hazard ratio (HR) and 95% CIs for the relationship between antidepressant discontinuation versus continuation and psychiatric emergency. Each matched pair constituted a separate stratum, and each stratum had its baseline hazard function. In the case of the remaining imbalance, we additionally adjusted for the imbalanced covariates [33]. To determine whether the risk of psychiatric emergency changed over time, we investigated the risk during pregnancy and the postpartum period separately, and the date at delivery was used to define the end of follow-up for pregnancy and the start of follow-up for postpartum period. Analyses were performed in Stata, version 15.0 (Stata, College Station, Texas, US).

## Ethical approval

The study was approved by the Danish Data Protection Agency. By Danish law, no informed consent is required for a register-based study on the basis of anonymized data.

## Results

Of 23,189 women included in the study, 4,368 (18.8%) discontinued antidepressants before pregnancy, 13,258 (57.2%) discontinued during pregnancy, and 5,563 (24.0%) continued antidepressant treatment throughout the entire pregnancy (see S1 Fig for an overview of timing of discontinuation). Women who continued antidepressants differed from women who discontinued antidepressants before or during pregnancy; for instance, women who continued treatment were more likely to have a psychiatric disorder diagnosis and an earlier onset of affective episode. For further characteristics of the entire unpaired study population, see S4 Table. The distributions of propensity score among antidepressant discontinuation and antidepressant group before matching are shown in S2 and S3 Figs.

In total, 2,669 women who discontinued antidepressants before pregnancy and 5,467 women who discontinued treatment during pregnancy were each matched to one woman who continued treatment (Fig 1). Propensity score matching provided a good covariate balance between groups except for, in the cohort of 2,669 matched pairs, women who discontinued antidepressants before pregnancy had shorter education duration and were less likely to receive TCAs/MAOIs in the 90 days before pregnancy, as assessed by standardized difference <0.1. In the cohort of 5,467 matched pairs, those who discontinued antidepressants during pregnancy had a shorter education duration and were younger at first affective disorder episode (Table 1).

### Antidepressant discontinuation before pregnancy and subsequent psychiatric emergency in the perinatal period

Of 2,669 matched pairs, 76 women who discontinued antidepressant treatment before pregnancy and 91 women who continued had a psychiatric emergency in the perinatal period (cumulative incidence = 2.9%, 95% CI: 2.3% to 3.6% versus 3.4%, 95% CI: 2.8% to 4.2%, respectively). Fig 2

**Table 1. Characteristics of the study population after propensity score matching.**

| Characteristics | Antidepressant discontinuation before pregnancy and its matched continuation group | | | Antidepressant discontinuation during pregnancy and its matched continuation group | | |
|---|---|---|---|---|---|---|
| | Discontinuation group (*N* = 2,669) | Continuation group (*N* = 2,669) | Absolute standardized differences | Discontinuation group (*N* = 5,467) | Continuation group (*N* = 5,467) | Absolute standardized differences |
| **Age at the index pregnancy** | | | | | | |
| 18 to 25 | 727 (27.2) | 695 (26.0) | 0.09 | 1,161 (21.2) | 1,174 (21.5) | 0.09 |
| 26 to 29 | 676 (25.3) | 758 (28.4) | | 1,416 (25.9) | 1,547 (28.3) | |
| 30 to 34 | 785 (29.4) | 808 (30.3) | | 1,761 (32.2) | 1,805 (33.0) | |
| ≥35 | 481 (18.0) | 408 (15.3) | | 1,129 (20.7) | 941 (17.2) | |
| **Parity** | | | | | | |
| First | 1,244 (46.6) | 1,230 (46.1) | 0.01 | 2,705 (49.5) | 2,660 (48.7) | 0.02 |
| Second and above | 1,425 (53.4) | 1,439 (53.9) | | 2,762 (50.5) | 2,807 (51.3) | |
| **Marital status** | | | | | | |
| Married or cohabiting | 1,856 (69.5) | 1,796 (67.3) | 0.00 | 4,104 (75.1) | 3,890 (71.2) | 0.02 |
| Single, divorced, or widowed | 813 (30.5) | 873 (32.7) | | 1,363 (24.9) | 1,577 (28.8) | |
| **Level of education** | | | | | | |
| Mandatory school comprising 9 school years | 960 (36.0) | 747 (28.0) | 0.21 | 1,688 (30.9) | 1,389 (25.4) | 0.15 |
| Above mandatory school | 1,630 (61.1) | 1,880 (70.4) | | 3,607 (66.0) | 3,969 (72.6) | |
| Unknown | 79 (3.0) | 42 (1.6) | | 172 (3.1) | 109 (2.0) | |
| **Age at first affective episode** | | | | | | |
| ≤18 | 338 (12.7) | 300 (11.2) | 0.08 | 930 (17.0) | 751 (13.7) | 0.11 |
| 19 to 25 | 1,152 (43.2) | 1,217 (45.6) | | 2,613 (47.8) | 2,581 (47.2) | |
| 26 to 29 | 623 (23.3) | 647 (24.2) | | 1,097 (20.1) | 1,243 (22.7) | |
| 30 to 34 | 425 (15.9) | 406 (15.2) | | 673 (12.3) | 729 (13.3) | |
| ≥35 | 131 (4.9) | 99 (3.7) | | 154 (2.8) | 163 (3.0) | |
| **History of suicide attempts before pregnancy** | 209 (7.8) | 207 (7.8) | 0.00 | 469 (8.6) | 476 (8.7) | 0.00 |
| **Psychiatric diagnosis before pregnancy** | | | | | | |
| Substance abuse disorder | 92 (3.4) | 89 (3.3) | 0.01 | 212 (3.9) | 216 (4.0) | 0.00 |
| Schizophrenia | 14 (0.5) | 20 (0.7) | 0.03 | 29 (0.5) | 30 (0.5) | 0.00 |
| Bipolar disorder | 19 (0.7) | 21 (0.8) | 0.01 | 112 (2.1) | 105 (1.9) | 0.01 |
| Depression | 519 (19.4) | 553 (20.7) | 0.03 | 1,654 (30.3) | 1,430 (26.2) | 0.09 |
| Other mood disorder | 30 (1.1) | 33 (1.2) | 0.01 | 112 (2.0) | 105 (1.9) | 0.01 |
| Neurotic, stress-related, and somatoform disorders | 660 (24.7) | 645 (24.2) | 0.01 | 1,946 (35.6) | 1,694 (31.0) | 0.10 |
| Personality disorders | 368 (13.8) | 377 (14.1) | 0.01 | 1,079 (19.7) | 938 (17.2) | 0.07 |
| Child onset psychiatric disorders | 67 (2.5) | 63 (2.4) | 0.01 | 149 (2.7) | 152 (2.8) | 0.00 |
| Other disorders | 283 (10.6) | 288 (10.8) | 0.01 | 815 (14.9) | 729 (13.3) | 0.05 |
| **Nonpsychiatric comorbidity scores** | | | | | | |
| 0 | 2,350 (88.0) | 2,352 (88.1) | 0.02 | 4,771 (87.3) | 4,768 (87.2) | 0.02 |
| 1 | 249 (9.3) | 256 (9.6) | | 541 (9.9) | 560 (10.2) | |
| ≥2 | 70 (2.6) | 61 (2.3) | | 155 (2.8) | 139 (2.5) | |
| **Number of previous psychiatric emergencies prior to 90 days before pregnancy** | | | | | | |
| 0 | 2,079 (77.9) | 2,057 (77.1) | 0.03 | 3,756 (68.7) | 3,944 (72.1) | 0.08 |

(*Continued*)

**Table 1.** (Continued)

| Characteristics | Antidepressant discontinuation before pregnancy and its matched continuation group | | | Antidepressant discontinuation during pregnancy and its matched continuation group | | |
|---|---|---|---|---|---|---|
| | Discontinuation group (N = 2,669) | Continuation group (N = 2,669) | Absolute standardized differences | Discontinuation group (N = 5,467) | Continuation group (N = 5,467) | Absolute standardized differences |
| 1 | 293 (11.0) | 305 (11.4) | | 792 (14.5) | 678 (12.4) | |
| 2 | 118 (4.4) | 116 (4.3) | | 383 (7.0) | 327 (6.0) | |
| 3 to 4 | 102 (3.8) | 116 (4.3) | | 278 (5.1) | 279 (5.1) | |
| ≥5 | 77 (2.9) | 75 (2.8) | | 258 (4.7) | 239 (4.4) | |
| **Psychiatric emergency in the 90 days before pregnancy** | 61 (2.3) | 63 (2.4) | 0.00 | 112 (2.1) | 113 (2.1) | 0.00 |
| **Classes of antidepressant treatment in the 90 days before pregnancy**[a] | | | | | | |
| SSRIs | 2,046 (76.7) | 1,932 (72.4) | 0.10 | 4,759 (87.0) | 4,579 (83.8) | 0.09 |
| SNRIs | 744 (27.9) | 772 (28.9) | 0.02 | 812 (14.9) | 948 (17.3) | 0.07 |
| TCAs or MAIOs | 91 (3.4) | 158 (5.9) | 0.12 | 141 (2.6) | 188 (3.4) | 0.05 |
| **Comedications in the 90 days before pregnancy** | | | | | | |
| Benzodiazepine | 273 (10.2) | 271 (10.2) | 0.00 | 585 (10.7) | 541 (9.9) | 0.03 |
| Anxiolytics | 5 (0.2) | 5 (0.2) | 0.00 | 13 (0.2) | 10 (0.2) | 0.01 |
| Antipsychotics | 157 (5.9) | 152 (5.7) | 0.01 | 503 (9.2) | 448 (8.2) | 0.04 |
| Opioid | 122 (4.6) | 131 (4.9) | 0.02 | 199 (3.6) | 211 (3.9) | 0.01 |
| Antiepileptics | 94 (3.5) | 111 (4.2) | 0.03 | 312 (5.7) | 292 (5.3) | 0.02 |
| Other hypnotics | 12 (0.5) | 7 (0.3) | 0.03 | 19 (0.4) | 20 (0.4) | 0.00 |
| **Calendar year at the index pregnancy** | | | | | | |
| 1995 to 2004 | 698 (26.2) | 747 (28.0) | 0.04 | 1,055 (19.3) | 1,055 (19.3) | 0.03 |
| 2005 to 2009 | 1,306 (48.9) | 1,257 (47.1) | | 2,853 (52.2) | 2,790 (51.0) | |
| 2011 to 2015 | 665 (24.9) | 665 (24.9) | | 1,559 (28.5) | 1,622 (29.7) | |

[a]The number of classes of antidepressant does not add up to the total number of women since some women received more than one class of antidepressants.

Values are numbers (%) unless stated otherwise.

MAOI, monoamine oxidase inhibitor; SNRI, serotonin–norepinephrine reuptake inhibitor; SSRI, selective serotonin reuptake inhibitor; TCA, tricyclic antidepressant.

shows the Kaplan–Meier curves for psychiatric emergency in the perinatal period by antidepressant discontinuation before pregnancy versus continuation during pregnancy. Antidepressant discontinuation before pregnancy was not significantly associated with increased psychiatric emergency risk (HR = 0.84, 95% CI: 0.61 to 1.16, $p$ = 0.298). The risk of psychiatric emergency associated with antidepressant discontinuation before pregnancy did not differ between the pregnancy and the postpartum periods ($p$-value for interaction = 0.410).

## Antidepressant discontinuation during pregnancy and subsequent psychiatric emergency in the perinatal period

Of 5,467 matched pairs, 202 women who discontinued antidepressants during pregnancy and 156 who continued had a psychiatric emergency in the perinatal period (cumulative incidence = 5.0%, 95% CI: 4.2% to 5.9% versus 3.7%, 95% CI: 3.1% to 4.5%). Fig 3 shows the Kaplan–Meier curves for psychiatric emergency in the perinatal period by antidepressant discontinuation versus continuation during pregnancy. The risk of psychiatric emergency among women who discontinued antidepressants during pregnancy was 1.25 (95% CI: 1.00% to

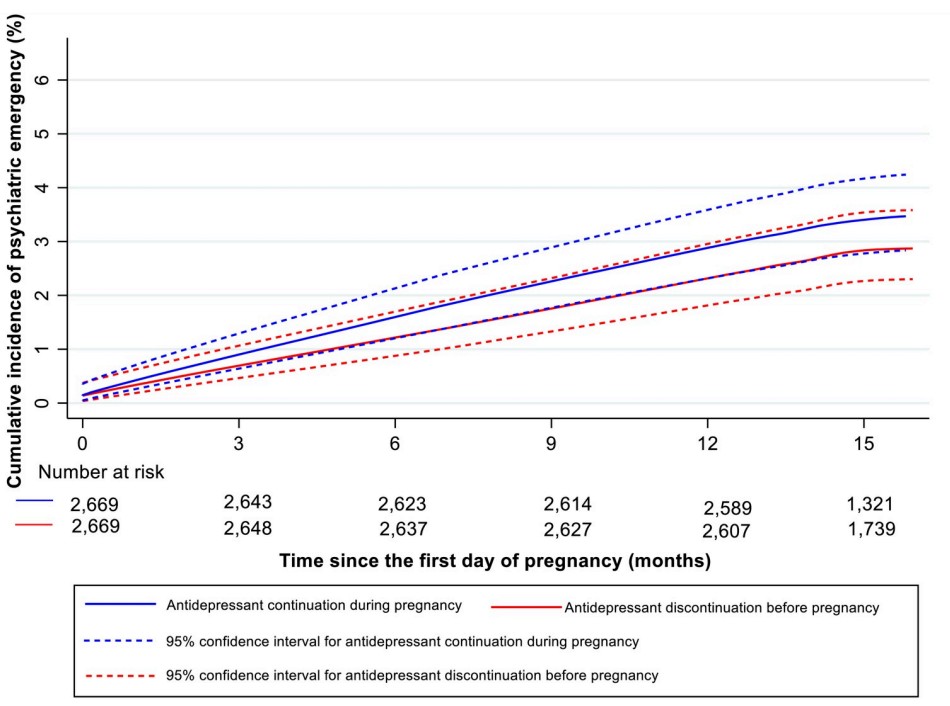

**Fig 2. Propensity score–matched smoothed Kaplan–Meier curves for psychiatric emergency since the start of pregnancy among antidepressant discontinuation before pregnancy group versus matched continuation during pregnancy group**[*]. [*]The cumulative incidence curve is smoothed to avoid personal identification according to the Data Protection Regulation in Denmark.

1.55%) times women who continued antidepressants ($p = 0.048$). The risk of psychiatric emergency associated with antidepressant discontinuation was increased during the observation time of pregnancy (HR = 1.52, 95% CI: 1.08 to 2.15, $p = 0.016$) but not during the postpartum period (HR = 1.09, 95% CI: 0.80 to 1.47, $p = 0.596$) (Table 2). There was no statistically significant interaction between antidepressant discontinuation during pregnancy and time of observation of psychiatric emergency ($p = 0.153$).

## Sensitivity analyses

When applying a less restrictive grace period of 30 days, in contrast to that of 14 days applied in the main analyses, findings were similar; however, the association between antidepressant discontinuation during pregnancy and psychiatric emergency was attenuated and not statistically significant (HR = 1.16, 95% CI: 0.96 to 1.41, $p = 0.156$), and antidepressant discontinuation before pregnancy was associated with a reduced risk of psychiatric emergency (HR = 0.66, 95% CI: 0.44 to 0.99, $p = 0.044$) (S5 Table). To investigate whether risk was affected by the class of antidepressants used in the 90 days before pregnancy, we limited our analyses to women who used SSRIs only. The associations remained similar, although they did not reach statistical significance: The HR was 0.74 (95% CI: 0.47 to 1.17, $p = 0.137$) for SSRI discontinuation before pregnancy and 1.16 (95% CI: 0.88 to 1.53, $p = 0.302$) for SSRI discontinuation during pregnancy, compared to women who continued SSRIs. To examine whether the associations differed by the severity of underlying episodes, we defined patients with severe disorders as being admitted to a psychiatric hospital, emergency room visits, or suicide attempts before the start of pregnancy and moderate disorders otherwise. The association between antidepressant

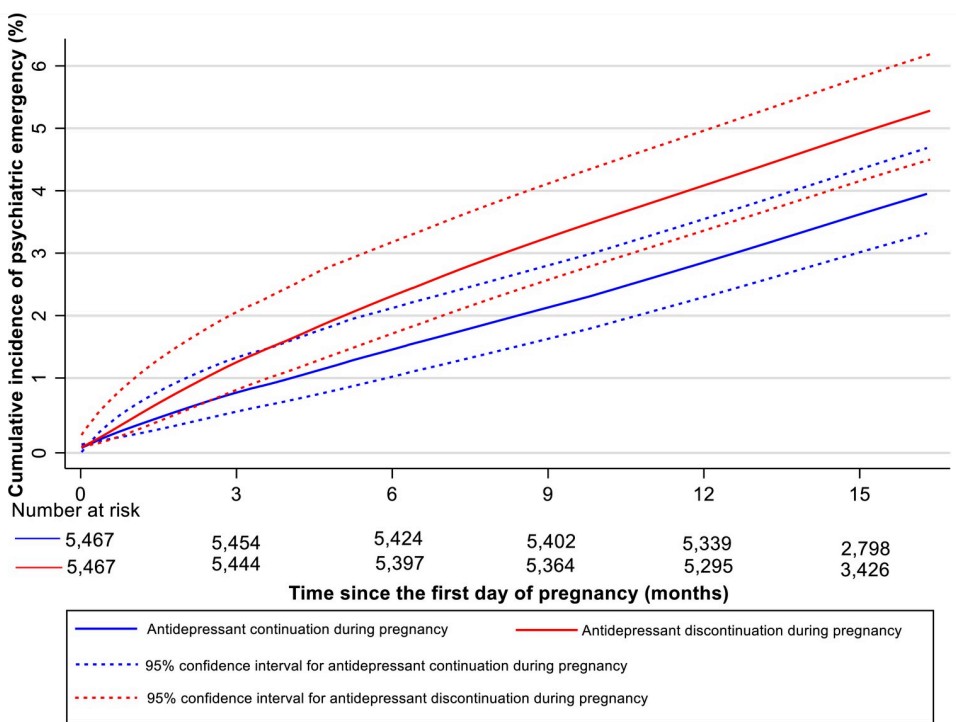

**Fig 3. Propensity score–matched smoothed Kaplan–Meier curves for psychiatric emergency since the start of pregnancy among antidepressant discontinuation during pregnancy group versus matched continuation during pregnancy group**\*. \*The cumulative incidence curve is smoothed to avoid personal identification according to the Data Protection Regulation in Denmark.

discontinuation during pregnancy and risk of psychiatric emergency was greater for women with severe disorders (HR = 2.68, 95% CI: 0.91 to 7.89, $p$ = 0.708) than those with moderate disorders (HR = 1.38, 95% CI: 1.00 to 1.89, $p$ = 0.050), although associations did not meet statistical significance and the CIs overlapped. Numbers were too small to give an accurate estimate for the matched cohort on antidepressant discontinuation before pregnancy. The results remained similar by including maternal age at pregnancy and age at the first affective disorder in the models using linear splines instead of categorical variables in the primary analysis (S6 Table).

## Discussion

Conducting a propensity score–matched cohort study, we found that antidepressant discontinuation during pregnancy was associated with an increased risk of psychiatric emergency during the perinatal period. There was no evidence of an association between antidepressant discontinuation before pregnancy and subsequent risk of psychiatric emergency.

### Antidepressant discontinuation during pregnancy and psychiatric emergency risk

Most studies on antidepressant treatment during pregnancy have so far centered on potential negative childhood outcomes, e.g., congenital malformations, neonatal persistent pulmonary hypertension, and neurodevelopment and psychiatric outcomes [6,7,34]. In contrast, the effectiveness of antidepressants in preventing psychiatric emergencies during pregnancy has not

**Table 2. Risk of psychiatric emergency associated with antidepressant discontinuation in propensity score–matched cohort analyses[a].**

| Matched groups according to the time of discontinuation of the exposed group | Antidepressant discontinuation group | | | | Antidepressant continuation group | | | | Unadjusted HRs (95% CI) | Adjusted HRs (95% CI)[b] | p-Values for adjusted analyses |
|---|---|---|---|---|---|---|---|---|---|---|---|
| | No. of women | No. of events | Person-years | Incidence/ 1,000 person-years | No. of women | No. of events | Person-years | Incidence/ 1,000 person-years | | | |
| **Antidepressant discontinuation prior to pregnancy** | 2,669 | 76 | 3,309.28 | 23.0 | 2,669 | 91 | 3,258.89 | 27.9 | 0.84 (0.62 to 1.15) | 0.84 (0.61 to 1.16) | 0.298 |
| During pregnancy | 2,669 | 40 | 2,002.38 | 20.0 | 2,669 | 52 | 1,959.36 | 26.5 | 0.75 (0.50 to 1.14) | 0.74 (0.47 to 1.15) | 0.175 |
| Within 6 months postpartum | 2,578 | 36 | 1,281.34 | 28.1 | 2,578 | 38 | 1,280.20 | 29.7 | 0.97 (0.61 to 1.56) | 0.98 (0.60 to 1.60) | 0.946 |
| **Antidepressant discontinuation during pregnancy** | 5,467 | 202 | 5,335.86 | 37.9 | 5,467 | 156 | 5,312.48 | 29.4 | 1.29 (1.05 to 1.59) | 1.25 (1.00 to 1.55) | 0.048 |
| During pregnancy | 5,467 | 92 | 2,677.95 | 34.4 | 5,467 | 58 | 2,632.62 | 22.0 | 1.59 (1.14 to 2.20) | 1.52 (1.08 to 2.15) | 0.016 |
| Within 6 months postpartum | 5,316 | 106 | 2,630.23 | 40.3 | 5,316 | 98 | 2,635.26 | 37.2 | 1.15 (0.85 to 1.54) | 1.09 (0.80 to 1.47) | 0.596 |

[a]The numbers of cases of psychiatric emergency occurred during pregnancy and within 6 months postpartum do not add up to the whole period since only matched individuals contribute to the analyses.

[b]Adjustment for imbalanced variables: level of education status and the use of TCAs or MAOIs in the 90 days before pregnancy for the estimate of antidepressant discontinuation before pregnancy and the level of education and age at first affective disorder for the estimate of antidepressant discontinuation during pregnancy.

CI, confidence interval; HR, hazard ratio; MAOI, monoamine oxidase inhibitor; TCA, tricyclic antidepressant.

received similar attention. A limited number of previous studies on relapse or worsening of symptoms after discontinuing antidepressants during pregnancy report conflicting findings [9–13]. In the current study, we found an increased risk of psychiatric emergency among women who discontinued antidepressants during pregnancy compared to those who continued treatment (HR = 1.25, 95% CI: 1.00 to 1.55). Although inconclusive due to limited numbers, the point estimate suggests that the increased risk of psychiatric emergencies following antidepressant discontinuation may be more pronounced among women with severe psychiatric disorders, in line with the recent meta-analysis [14]. We note that our observed effect size is smaller than previously reported 5.0 (95% CI: 2.8 to 9.1) or 8.1 (95% CI: 2.4 to 27.0) [9,10], which most likely reflects that previous studies were performed on small and selected study populations, applying different methods, or outcome measurements.

It could be argued that women on continuous treatment may interact more regularly with the healthcare system and potentially experience positive effects of routine monitoring. We specifically note that 95% of the women who discontinued antidepressants did not experience our defined outcome, psychiatric emergency in the perinatal period, which could be seen as a reassuring message for women contemplating discontinuing their treatment. We also noted that the increased risk associated with antidepressant discontinuation was more pronounced in pregnancy (HR = 1.52, 95% CI: 1.08 to 2.15) than in the postpartum period (HR = 1.09, 95% CI: 0.80 to 1.47). We speculate that women who discontinue antidepressants during pregnancy may restart their medication treatment, in particular, after childbirth [35] and thus reduce the risk of a psychiatric emergency. Notably, the 95% CIs overlapped, and the difference in the risk noted should be interpreted with caution.

Furthermore, our results should be interpreted with caution because psychiatric emergency is a severe outcome. In contrast, a larger proportion of these women may have a recurrent

episode that does not necessarily lead to emergency room visits or admissions. It would be interesting to investigate less severe outcomes such as worsening of symptoms or affective instability in future prospective clinical studies. Moreover, future efforts should also be made to identify which women are at low or high risk of psychiatric emergency to guide more nuanced treatment recommendations (e.g., preventive nonpharmacological treatment) [36].

## Antidepressant discontinuation before pregnancy and psychiatric emergency risk

To the best of our knowledge, no previous studies have investigated the risk of psychiatric emergency in women who discontinued antidepressants before pregnancy. We found that antidepressant discontinuation in 90 days before pregnancy was not associated with an increased risk of psychiatric emergency in the primary analysis, suggesting that women who discontinued antidepressants before pregnancy may differ from those who discontinued treatment during pregnancy. Although not investigated in this study, we speculate that women who discontinue antidepressants prior to conception do this as part of pregnancy planning or represent a selected population of women, which may not be directly comparable to women who discontinue during pregnancy. Discontinuation during pregnancy might be associated with an unplanned pregnancy, and women might more often stop immediately instead of gradually tapering off their medication, which is an identified risk factor for worsening of mood [37]. A further reduced risk was observed when applying a less restrictive grace period of 30 days, in contrast to 14 days, although the 95% CIs overlapped. One explanation for this particular difference could relate to differences in pregnancy planning between the 2 groups; however, any explanations are at this point highly speculative, as we do not have information on reasons for starting and stopping medication use.

## Decisions on whether or not to continue antidepressants during pregnancy

Our findings add to the limited evidence on the efficacy of continued antidepressant treatment during pregnancy to prevent a psychiatric emergency, which has direct clinical relevance. We acknowledge that decisions on antidepressant continuation depend not only on evidence-based knowledge of the benefits and risks but also on the women's perceptions of risk, values, and treatment preferences [38]. In our study, over 50% of women discontinued antidepressants during pregnancy. Although we do not know the reasons these women discontinued antidepressants, we speculate that perception of fetal risk may be the main reason [6,39,40], despite the fact that evidence on the potential effect of antidepressant exposure in utero is inconclusive and the absolute potential risk is low [6,34,40]. Available data suggest that antidepressants are not major teratogens [34], but the evidence on longer-term outcomes in offspring is inconclusive [7]. Regardless, the provision of evidence-based risk–benefit information through counseling can help women attain more nuanced risk perceptions.

## Strengths and limitations

Our study exhibits several strengths. It was based on a representative cohort of women from the entire Danish population and included all pregnant women treated with antidepressants in the 90 days prior to pregnancy. The linkage of several national registers enabled us to control, at least partially, for confounding by the underlying condition through adjustment for several covariates which may be a proxy of disease severity, such as previous psychiatric diagnosis, psychiatric emergency, and concomitant use of medications in the 90 days prior to pregnancy. The use of propensity score matching enabled us to achieve comparability between

discontinuation and continuation groups and thus get a more accurate estimate of the effectiveness of continuous antidepressant treatment on psychiatric emergencies in the perinatal period.

Our study also has some limitations. First, some women who filled the prescriptions may not take antidepressants, and we might misclassify them as continued treatment. However, adherence to antidepressant treatment during pregnancy is high in Denmark [41]. Therefore, misclassification is likely to be limited. Moreover, we did not have accurate information on the duration of antidepressant discontinuation. We estimated this based on number of defined daily doses; however, as dosage may vary between individuals, we would have misclassified antidepressant continuation versus discontinuation status for some individuals. This would have biased the association of psychiatric emergency with antidepressant discontinuation toward the null. Similarly, we may have misclassified the time of discontinuation, making the associations with discontinuation before pregnancy and during pregnancy more similar. Second, information on reasons for discontinuing, diagnoses from general practitioners, and non-pharmaceutical treatment is not available in the registers. We further do not know the indication for initiating antidepressant treatment. The national registers provide detailed information on hospital and pharmaceutical treatment, and we tried to control for the unmeasured disorder severity by using propensity score matching. Nonetheless, we have no data on symptom severity and detailed baseline symptoms, leading to residual confounding. Third, some women were excluded from the matched cohort when we matched women who continued treatment during pregnancy to those who discontinued antidepressants before pregnancy.

## Conclusions

A substantial proportion of women discontinue antidepressant treatment around the time of conception. This large propensity score–matched cohort study suggests that discontinuing antidepressants specifically during pregnancy is associated with an increased risk of psychiatric emergency, compared to continuous antidepressant treatment without interruption. However, we observed that the absolute risk difference of psychiatric emergency is low (cumulative incidence of 5.0% in women who discontinue versus 3.7% in women who continue antidepressants). While causality cannot be determined from this study, if the association we observed indicates a causal relationship, continuing antidepressant treatment across pregnancy may be effective in reducing psychiatric emergency risk.

## Supporting information

**S1 STROBE Checklist. Checklist of items that should be included in reports of cohort studies.** STROBE, Strengthening the Reporting of Observational Studies in Epidemiology.
(PDF)

**S1 Text. Detailed description of Danish national registers used in this study.**
(PDF)

**S2 Text. Charlson comorbidity index.**
(PDF)

**S3 Text. Definition of suicide attempts.**
(PDF)

**S1 Table. The ICD-8 or ICD-10 codes for subgroup diagnosis of psychiatric comorbidities.**
(PDF)

**S2 Table. Classes of antidepressant treatment during pregnancy.**
(PDF)

**S3 Table. ATC codes for comedications in the 90 days prior to pregnancy.** ATC, Anatomical Therapeutic Chemical.
(PDF)

**S4 Table. Characteristics of the study population before propensity score matching.**
(PDF)

**S5 Table. Risk of psychiatric emergency associated with antidepressant discontinuation in propensity score matched cohort analyses using a less restrictive 30-day grace period.**
(PDF)

**S6 Table. Risk of psychiatric emergency associated with antidepressant discontinuation in propensity score–matched cohort analyses including age at pregnancy and age at first affective disorders as linear splines.**
(PDF)

**S1 Fig. The distribution of time of antidepressant discontinuation.**
(TIFF)

**S2 Fig. The distribution of propensity score of discontinuing antidepressants before pregnancy before matching.**
(TIFF)

**S3 Fig. The distribution of propensity score of discontinuing antidepressants during pregnancy before matching.**
(TIFF)

## Author Contributions

**Conceptualization:** Xiaoqin Liu, Esben Agerbo, Veerle Bergink, Trine Munk-Olsen.

**Data curation:** Xiaoqin Liu, Esben Agerbo.

**Formal analysis:** Xiaoqin Liu.

**Funding acquisition:** Xiaoqin Liu, Nina Molenaar, Veerle Bergink, Trine Munk-Olsen.

**Investigation:** Xiaoqin Liu, Nina Molenaar, Esben Agerbo, Natalie C. Momen, Anna-Sophie Rommel, Angela Lupattelli, Veerle Bergink, Trine Munk-Olsen.

**Methodology:** Xiaoqin Liu, Nina Molenaar, Esben Agerbo, Natalie C. Momen, Anna-Sophie Rommel, Angela Lupattelli, Veerle Bergink, Trine Munk-Olsen.

**Project administration:** Xiaoqin Liu, Nina Molenaar.

**Resources:** Xiaoqin Liu, Esben Agerbo, Trine Munk-Olsen.

**Software:** Xiaoqin Liu.

**Supervision:** Veerle Bergink, Trine Munk-Olsen.

**Validation:** Xiaoqin Liu, Esben Agerbo, Angela Lupattelli, Trine Munk-Olsen.

**Visualization:** Xiaoqin Liu.

**Writing – original draft:** Xiaoqin Liu.

**Writing – review & editing:** Xiaoqin Liu, Nina Molenaar, Esben Agerbo, Natalie C. Momen, Anna-Sophie Rommel, Angela Lupattelli, Veerle Berdink, Trine Munk-Olsen.

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
