## [Editor Report · Decision Letter 0]

9 Mar 2021

Dear Dr Liu, 

Thank you for submitting your manuscript entitled "Antidepressant discontinuation before or during pregnancy and risk of psychiatric emergency: a population-based propensity score-matched cohort study" for consideration by PLOS Medicine.

Your manuscript has now been evaluated by the PLOS Medicine editorial staff as well as by an academic editor with relevant expertise and I am writing to let you know that we would like to send your submission out for external peer review.

Kind regards,

Dr Raffaella Bosurgi

Executive Editor 

PLOS Medicine

---

## [Decision Letter · Decision Letter 1]

20 Sep 2021

Dear Dr. Liu,

Thank you very much for submitting your manuscript "Antidepressant discontinuation before or during pregnancy and risk of psychiatric emergency: a population-based propensity score-matched cohort study" (PMEDICINE-D-21-01037R1) for consideration at PLOS Medicine. 

Your paper was evaluated by a senior editor and discussed among all the editors here. It was also discussed with an academic editor with relevant expertise, and sent to four independent reviewers, including a statistical reviewer. The reviews are appended at the bottom of this email and any accompanying reviewer attachments can be seen via the link below:

[LINK]

The reviewers have pointed out some significant concerns, I am afraid that we will not be able to accept the manuscript for publication in the journal in its current form, but we will consider a revised version that addresses the reviewers' and editors' comments. Obviously we cannot make any decision about publication until we have seen the revised manuscript and your response, and we plan to seek re-review by one or more of the reviewers. 

We expect to receive your revised manuscript by Oct 11 2021 11:59PM. Please email us (plosmedicine@plos.org) if you have any questions or concerns.

We look forward to receiving your revised manuscript. 

Sincerely,

Caitlin Moyer, PhD

Associate Editor

PLOS Medicine

plosmedicine.org

1. We note that there are major concerns raised by the reviewers, especially reviewer 3, relating to the validity of the analyses. Please do respond to and address the comments of all the reviewers.

2. Data availability statement: The Data Availability Statement (DAS) requires revision. For each data source used in your study: 

3. Throughout: Please include line numbers with the revised version.

4. Abstract: Please combine the Methods and Findings sections into one section, “Methods and findings”.

5. Abstract: Methods and Findings: Please include the population and setting, number of participants, and years during which the study took place.

6. Abstract: Methods and Findings: Please include the important dependent variables that are adjusted for in the analyses.

7. Abstract: Methods and Findings: Please quantify the main results with both 95% CIs and p values.

8. Abstract: Methods and Findings: In the last sentence of the Abstract Methods and Findings section, please describe the main limitation(s) of the study's methodology.

9. Throughout: Please use square brackets for in-text citations, placed before the sentence punctuation. Where multiple references are included within brackets, please do not include spaces.

10. Methods: Please ensure that the study is reported according to the STROBE guideline, and include the completed STROBE checklist as Supporting Information. Please add the following statement, or similar, to the Methods: "This study is reported as per the Strengthening the Reporting of Observational Studies in Epidemiology (STROBE) guideline (S1 Checklist)."

11. Methods: Did your study have a prospective protocol or analysis plan? Please state this (either way) early in the Methods section.

12. Methods: Please provide more detail on the Danish nationwide registers used for the cohort study.

13. Methods: Please comment on whether it is possible to examine recurrent illness that may not necessarily present in the ER/ hospital perhaps due to being less severe relative to those illnesses that may present in the ER/hospital.

14. Results: Please present both 95% CIs and p values for all results described in the text; for example, for the association of antibdepressant discontinuation with psychiatric emergency risk.

15. Results: Please note where the results to support this statement are presented: “The risk of psychiatric emergency associated with antidepressant discontinuation before pregnancy did not differ between the pregnancy and the postpartum periods” (please note where the results for the interaction test can be found).

16. Results: Please clarify for the analysis of those who discontinued medication during pregnancy, that the interaction between time of observing the outcome of psychiatric emergency (during pregnancy or postpartum) did not reach significance. This seems to suggest that the different risk noted should be interpreted with caution given the lack of an interaction.

17. Results: “When applying a less restrictive grace period of 30-days, in contrast to that of 14-days applied in the main analyses, findings were similar; however, associations were attenuated (HR=1.16, 95%CI:0.96–1.41)...” Please clarify which associations are being referred to here.

18. Results: Sensitivity analysis: Please include in the discussion some interpretation on why the association between discontinuation during pregnancy and psychiatric emergency might reverse direction with the 30 day rather than 14 day grace period. Please note in the text the sensitivity analyses for which associations observed in main analyses were attenuated/statistical significance no longer apparent.

19. Discussion: Please present and organize the Discussion as follows: a short, clear summary of the article's findings; what the study adds to existing research and where and why the results may differ from previous research; strengths and limitations of the study; implications and next steps for research, clinical practice, and/or public policy; one-paragraph conclusion.

20. Discussion: “No previous studies have investigated risk of psychiatric emergency in women who discontinued antidepressants before pregnancy.” Please qualify this and any similar statements with “To the best of our knowledge…” or similar.

21. Conclusions: “This large propensity score-matched cohort study suggests that discontinuing antidepressants specifically during pregnancy increases risk of psychiatric emergency…” Please avoid the use of language implying causality, here and throughout the manuscript.

22. References: Please use the "Vancouver" style for reference formatting, and see our website for other reference guidelines https://journals.plos.org/plosmedicine/s/submission-guidelines#loc-references

23. Table 2 and eTable 5: Please also provide the results of the unadjusted analyses.

Comments from the reviewers:

Reviewer #1: I confine my remarks to statistical aspects of this paper. These were generally very well done, and I almost voted to accept, but I have a couple of suggestions that the authors might implement, especially if they have to make other revisions.

p. 2 Results - there was, in fact, some evidence of *lower* risk among those who discontinued before pregnancy.

p. 4 It would be good to give effect size estimates of the other studies. 

 The results of the meta-analysis are very odd. There must have been huge heterogeneity to get such a large confidence interval. The authors might discuss this. 

p, 7 Although the results of the propensity analysis were very good, they might be even better if the authors used splines of age as a continuous variable, rather than catgegorizing it.

Peter Flom

Reviewer #2: Overall, this is interesting, important, rigorous work. I appreciate the detailed description of the cohort and matching. Outcomes (emergency visit or admission) are somewhat narrow, as acknowledged by authors, in that this probably does not capture the spectrum of the burden of recurrent illness during pregnancy/postpartum and many (perhaps most) women are probably not making a decision to continue or discontinue medication during pregnancy based on those outcomes, unless they have previously experienced those events.

Several minor comments.

Abstract: It might be more accurate to say that there is limited evidence on continuing vs discontinuing antidepressants in pregnancy. 

Introduction:

2-8% of pregnant women in Europe and 8-13% in the US receive an antidepressant prescription at some time during their pregnancy (versus continuous use).

Untreated/undertreated depression and anxiety should also be avoided to prevent maternal/fetal/infant death.

Methods:

If 41.3% had a psychiatric diagnosis, what were the indications for antidepressant use in the rest of the cohort? This is brought up in methods and again the discussion. Would be helpful to understand if having a documented psychiatric diagnosis indicates more severe illness.

Can you comment on if there was a group of women discontinued antidepressants before or during pregnancy and then restarted them during pregnancy or during postpartum? 

Reviewer #3: The authors try to address the exceedingly important question of navigating the risk benefit decision with respect to antidepressant use during pregnancy: to continue or to discontinue AD treatment given the relative knowns and unknowns associated with fetal exposure to this class of medication . The analytic exercise performed is profoundly flawed and does not contribute to enhanced ability to navigate the clinical dilemma . Indeed , there are some conflicting data on the question of benefit of antidepressant continuation during pregnancy . And yet this paper looks at an outcome variable - psychiatric emergency /ER visit /psych admission with respect to AD discontinuation that does not inform in any way . Do the authors make the argument that they have the ability to adjust for confounding based on the data in hand from an administrative database ? The variables in the propensity score model are appropriate but the reader can have sparse confidence that data reflecting accurately those variables is obtainable. An even greater limitation of this analysis is the failure to know the status of subjects in each cohort at baseline : were these women euthymic at baseline , before they discontinued treatment ? What value is the analysis if we do not know how women were clinically before looking at the outcome of interest . This may also be relevant as treatment of depression before , during and after pregnancy has certainly evolved across the interval of interest noted in the paper - 1996-2016.

Two other smaller issues : First , the reference to adverse effects of AD exposure during pregnancy being associated with persistent pulmonary hypertension of the newborn is puzzling as the concern has been put to rest in multiple reports . And yet these authors note as an outcome of concern with respect to fetal exposure to AD. Second , there is no discussion as to why there is no observation of increased frequency of the outcome postpartum which would be expected . 

There is a responsibility as studies get conducted in this area to make an effort to refine the risk benefit decision regarding the use of AD before, during and after pregnancy in reproductive age women . This manuscript does not advance the effort .

Reviewer #4: This manuscript is a cohort study that examines the effects of antidepressant discontinuation before and during pregnancy. The authors report an increased risk of psychiatric emergency in women who discontinue antidepressants during pregnancy. The risk is smaller than has been previously observed in the literature but clinically significant. There are several strengths of this manuscript including large sample size, transparent and thoughtful methodology, use of propensity score matching and a well-written paper. There are some minor critiques that I have of this manuscript that could improve it with revision but overall it is an important and meaningful work.

(1) Please define which medications are defined as antidepressants in the methods section. It is clear this is SSRI, SNRI, TCA and MAOI from table 1 but this should be mentioned in the text of the manuscript. 

(2) I wish the authors discussed more extensively the evidence (or lack thereof) of congenital malformations with antidepressant exposure. This is important when discussing the risk/benefits of antidepressant discontinuation during pregnancy.

(3) Although the absolute risk of psychiatric emergency is smaller than previous studies, I am not sure I would necessarily qualify it as that small of an absolute risk (not to mention it is not a small relative risk). The NNT seems to be around 77 and is probably much higher in individuals with mdoerate-to-severe illness or at the higher end of your propensity score.

(4) I also wish the authors commented on the high rate of women that discontinued antidepressant medication during pregnancy. it would be useful to examine whether there was any relationship between propensity score and decision to discontinue medication during pregnancy.

[LINK]

---

## [Decision Letter · Decision Letter 2]

3 Dec 2021

Dear Dr. Liu,

Thank you very much for re-submitting your manuscript "Antidepressant discontinuation before or during pregnancy and risk of psychiatric emergency: a population-based propensity score-matched cohort study" (PMEDICINE-D-21-01037R2) for review by PLOS Medicine.

I have discussed the paper with my colleagues and the academic editor and it was also seen again by three reviewers. I am pleased to say that provided the remaining editorial and production issues are dealt with we are planning to accept the paper for publication in the journal.

[LINK]

We look forward to receiving the revised manuscript by Dec 10 2021 11:59PM.   

Sincerely,

Caitlin Moyer, PhD

Associate Editor 

PLOS Medicine

plosmedicine.org

Requests from Editors:

1. Title: Please capitalize the first word of the subtitle: “Antidepressant discontinuation before or during pregnancy and risk of psychiatric emergency: A population-based propensity score-matched cohort study” and please update this in the manuscript submission system.

2. Data availability statement: Thank you for providing a statement regarding the availability of the data underlying your study. Please add the weblinks and contact email address(es) for the registers used in the study (e.g. https://econ.au.dk/the-national-centre-for-register-based-research/danish-registers/the-danish-civil-registration-system-cpr;
https://econ.au.dk/the-national-centre-for-register-based-research/danish-registers/the-national-patient-register;
https://econ.au.dk/the-national-centre-for-register-based-research/danish-registers/the-danish-psychiatric-central-research-register;
https://econ.au.dk/the-national-centre-for-register-based-research/danish-registers/the-medical-register-of-births-and-deaths if these links are applicable. Please also provide a link and contact information for the Danish National Prescription Registry).

3. Abstract: Methods: Line 34: Please revise to “We carried out a propensity score-matched cohort study of women with singleton live births between…” or similar. Please also note the setting of the study (Denmark) in the first sentence.

4. Abstract: Methods: Line 35-36: Please broadly define the categories of “antidepressants” considered.

5. Abstract: Methods and Findings: Please avoid the use of italics for emphasis.

6. Abstract: Methods and Findings: Please also provide the cumulative incidence for those who discontinued before pregnancy compared to those who did not.

7. Abstract: Conclusions: Please revise to: “In this study, we found that discontinuing antidepressant medication during pregnancy (but not before) is associated with an apparent increased risk of psychiatric emergency compared to continuing treatment across pregnancy.”

8. Author summary: What do these findings mean? We suggest revising to: “Our findings add to the limited evidence on the relationship between continued antidepressant treatment during pregnancy and perinatal psychiatric emergency.”

9. Introduction: Line 97: Please avoid the use of italics for emphasis, here and throughout the text.

10. Methods: Line 105-106: Thank you for your response indicating that there was no prospectively developed protocol for the study. Please include a sentence noting this in the main text, early on in the Methods section, similar to: “There is no documented analysis plan associated with the study. We planned our analyses on a detailed discussion among the authors and agreed on an outline for how the work would be done, as described in the Methods. Changes to the pre-planned analyses include…” and please specifically note the changes to the analyses that were implemented post-hoc, e.g. in response to peer reviewer comments. We ask that you please indicate in the text: (1) the specific hypotheses you intended to test, (2) the analytical methods by which you planned to test them, (3) the analyses you actually performed, and (4) when reported analyses differ from those that were planned, transparent explanations for differences that affect the reliability of the study's results. If a reported analysis was performed based on an interesting but unanticipated pattern in the data, please be clear that the analysis was data-driven.

11. Methods: Line 110-111: Thank you for including the detailed information in the Supporting Information files. In the main text, please do at least mention the names of the registers (e.g. the register from which pregnancies and births were identified, etc).

12. Methods: Line 160: Please briefly explain “mandatory school” in terms of what this indicates.

13. Methods: Line 165-166: Please either list the 19 comorbid conditions here, or provide a supporting information file with the list of conditions.

14. Methods: Line 216: Please list the imbalance covariates adjusted for in the adjusted analyses.

15. Results: Line 249-250: Please add “significantly” to the sentence: “Antidepressant discontinuation before pregnancy was not significantly associated with…”

16. Results: Line 251-253: Please provide the results (for the interaction) in the text that support this statement: “The risk of psychiatric emergency associated with antidepressant discontinuation before pregnancy did not differ between the pregnancy and the postpartum periods.”

17. Results: Line 257: Please clarify to “A total of 202 women who discontinued during pregnancy…” or similar.

18. Results: Line 263: Please clarify to “The risk of psychiatric emergency associated with antidepressant discontinuation during pregnancy…” if accurate.

19. Results: Line 265: We suggest revising to: “There was no statistically significant interaction between antidepressant discontinuation during pregnancy and time of observation of psychiatric emergency (p=0.153).” or similar.

20. Results: Lines 277-279: We suggest revising to “The associations remained similar, though they did not reach statistical significance: the HR was 0.74 (95%CI: 0.47–1.17, p=0.137) for SSRI discontinuation before pregnancy and 1.16 (95%CI: 0.88–1.53, p=0.302) for SSRI discontinuation during pregnancy, compared to women who continued SSRIs.” or similar.

21. Results: Line 281: Please clarify “having admission” in this sentence.

22. Results: Line 282-285: We suggest revising to: “The association between antidepressant discontinuation during pregnancy and risk of psychiatric emergency was greater for women with severe disorders (HR=2.68, 95%CI: 0.91–7.89, p=0.708) than those with moderate disorders (HR=1.38, 95%CI: 1.00–1.89, p=0.050), although associations did not meet statistical significance and the confidence intervals overlapped.”

23. Discussion: Lines 292-294: We suggest revising to: “Conducting a propensity score-matched cohort study, we found that antidepressant discontinuation during pregnancy was associated with an increased risk of psychiatric emergency during the perinatal period.” as there was no significant interaction between psychiatric emergency during pregnancy and the 6 months following pregnancy.

24. Discussion: Line 300: Here and throughout, where multiple references are indicated within brackets, please do not include spaces [6,7,34].

25. Discussion: Lines 306-308: Please revise to “...the point estimate suggests that the increased risk of psychiatric emergencies following antidepressant discontinuation may be more pronounced among women with severe psychiatric disorders…”

26. Discussion: Lines 347-349: It would be helpful to further clarify this sentence, in terms of why the less-restrictive grace period would lead to a reversal of the direction of the association with the 14 day grace period. “It is probable that women who get pregnant after discontinuing antidepressants 30 days may have planned their pregnancy even earlier than those who get pregnant 14 days after discontinuation.”

27. Discussion: Line 359: We suggest removing “(and will likely remain)” from the sentence.

28. Discussion: Line 377: Please use “adherence” rather than “compliance” here and throughout the manuscript.

29. Discussion: Line 399-403: We suggest revising to: “However, we observed that the absolute risk difference of psychiatric emergency is low (cumulative incidence of 5.0% in women who discontinue versus 3.7% in women who continue antidepressants). While causality cannot be determined from this study, if the association we observed indicates a causal relationship, continuing antidepressant treatment across pregnancy may be effective in reducing psychiatric emergency risk.”

30. Line 405: Please remove the “Competing Interests” section from the main text, and please ensure all information is complete and accurate in the Competing Interests section of the manuscript submission system.

31. Lines 405-426: Please remove the Funding, Contributors, and Data Sharing sections from the main text and please ensure all information is completely and accurately entered into the relevant sections of the manuscript submission system.

32. References: Please use the "Vancouver" style for reference formatting, and see our website for other reference guidelines https://journals.plos.org/plosmedicine/s/submission-guidelines#loc-references

Please avoid the use of italics.

33. Figure 1: Please revise to “680,312 women aged 18 years or older” in the first box.

34. Figure 2: For comparison, could the curves also be shown for those who discontinued antidepressant prior to pregnancy? Please provide the number at risk for each relevant time interval.

35. Table 1: Please define all abbreviations used in the table. Please include a definition of “mandatory school” in the legend.

36. Supporting Information file: Please submit a clean version of the file (without tracked changes). Please include both titles and legends for all figures and tables.

37. Supporting Information S2: Please provide a citation or other quantitative support for the sentence “The information is of very high quality.”

38. S1 Figure: Please provide a descriptive legend for the figure, and please provide more information for the y-axis label.

39. S4 Table: Please define all abbreviations used in the table. Please include a definition of “mandatory school” in the legend.

40. STROBE Checklist: Please provide the checklist as a separate file.

Comments from Reviewers:

Reviewer #1: The authors have addressed my concerns and I now recommend publication.

Peter Flom

Reviewer #2: I am satisfied with the revisions and have no additional comments to offer. The work remains important and although it is not possible to control for all confounders, etc, this does add to the field and could be used in helping clinicians/patients determine risk of continuing vs discontinuing medications preconception and during pregnancy.

Reviewer #4: The reviewers responded adequately to my critiques. I believe many of the important criticisms raised by other reviewers have also been adequately addressed in the limitations section and are well-acknowledged by the authors in the revised manuscript.

[LINK]

---

## [Editor Report · Decision Letter 3]

20 Dec 2021

Dear Dr Liu, 

On behalf of my colleagues and the Academic Editor, Mark Tomlinson, I am pleased to inform you that we have agreed to publish your manuscript "Antidepressant discontinuation before or during pregnancy and risk of psychiatric emergency: A population-based propensity score-matched cohort study" (PMEDICINE-D-21-01037R3) in PLOS Medicine.

Please also address the following editorial points:

1. Title: We suggest including the study setting: “Antidepressant discontinuation before or during pregnancy and risk of psychiatric emergency in Denmark: A population-based propensity score-matched cohort study” or similar.

2. Supplementary material. In the published article, supporting information files are accessed only through a hyperlink attached to the captions. For this reason, please list captions at the end of your manuscript file. You may include a caption within the supporting information file itself, as long as that caption is also provided in the manuscript file. Please do not submit a separate caption file. The supporting information file name and number are required in a caption, and we highly recommend including a one-line title and a caption as well. We also suggest including each supporting information figure and table as a file.

3. Table 2, S5 Table, S6 Table: Please indicate whether the p values correspond to the unadjusted or adjusted analyses.

4. Acknowledgements: Please note that it is not necessary to remove the acknowledgements section from the manuscript, if including one is preferred, as long as complete funding information is included in the relevant “Funding” section within the manuscript submission system.

PRESS

Sincerely, 

Caitlin Moyer, Ph.D. 

Associate Editor 

PLOS Medicine